# Metal-Responsive Transcription Factors Co-Regulate Anti-Sigma Factor (Rsd) and Ribosome Dimerization Factor Expression

**DOI:** 10.3390/ijms24054717

**Published:** 2023-03-01

**Authors:** Hideji Yoshida, Tomohiro Shimada, Akira Ishihama

**Affiliations:** 1Department of Physics, Osaka Medical and Pharmaceutical University, Takatsuki 569-8686, Osaka, Japan; 2School of Agriculture, Meiji University, Kawasaki 214-8571, Kanagawa, Japan; 3Micro-Nano Technology Research Center, Hosei University, Koganei 184-0003, Tokyo, Japan

**Keywords:** metal-responsive transcription factor, transcriptional regulation, translational regulation, Rsd, RMF, 100S ribosome

## Abstract

Bacteria exposed to stress survive by regulating the expression of several genes at the transcriptional and translational levels. For instance, in *Escherichia coli*, when growth is arrested in response to stress, such as nutrient starvation, the anti-sigma factor Rsd is expressed to inactivate the global regulator RpoD and activate the sigma factor RpoS. However, ribosome modulation factor (RMF) expressed in response to growth arrest binds to 70S ribosomes to form inactive 100S ribosomes and inhibit translational activity. Moreover, stress due to fluctuations in the concentration of metal ions essential for various intracellular pathways is regulated by a homeostatic mechanism involving metal-responsive transcription factors (TFs). Therefore, in this study, we examined the binding of a few metal-responsive TFs to the promoter regions of *rsd* and *rmf* through promoter-specific TF screening and studied the effects of these TFs on the expression of *rsd* and *rmf* in each TF gene-deficient *E. coli* strain through quantitative PCR, Western blot imaging, and 100S ribosome formation analysis. Our results suggest that several metal-responsive TFs (CueR, Fur, KdpE, MntR, NhaR, PhoP, ZntR, and ZraR) and metal ions (Cu^2+^, Fe^2+^, K^+^, Mn^2+^, Na^+^, Mg^2+^, and Zn^2+^) influence *rsd* and *rmf* gene expression while regulating transcriptional and translational activities.

## 1. Introduction

When an organism is exposed to stress, the expression of many genes is regulated at transcriptional and translational levels. For instance, in Gram-negative bacteria, such as *Escherichia coli*, the sigma factor RpoD binds to RNA polymerase under favorable growth conditions and serves as a basic transcription mechanism [1]. However, when growth is arrested in response to stress, such as nutrient starvation, the anti-sigma factor Rsd is expressed and binds to the global regulator RpoD, which becomes inactivated [2]. In addition, the sigma factor RpoS binds to RNA polymerase and expresses stationary phase-specific genes [3,4], allowing *E. coli* to regulate the transcription of stress-responsive genes. However, at the translational level, ribosome modulation factor (RMF), which is expressed in response to growth arrest, binds to 70S ribosomes to form inactive 100S ribosomes (dimers of 70S ribosomes), which regulate translational activity [5,6]. It is worth noting that these transcriptional and translational regulations occur simultaneously under stress, suggesting that several stress-responsive transcription factors (TFs) are involved in the expression of *rsd* and *rmf* genes. Therefore, elucidating the mechanisms of *rsd* and *rmf* gene expression will be critical to understanding bacterial survival strategies.

In a previous study, we used a promoter-specific TF (PS-TF) screening system to identify TFs that regulate *rsd* and *rmf* gene expressions by comprehensively examining the binding of approximately 200 *E. coli* TFs to the promoter regions of *rsd* and *rmf* genes [7]. The results revealed that multiple TFs bind to the promoter regions of *rsd*, which regulates transcription, and *rmf*, which regulates translation. Thus, we reported in the study that transcription and translation are simultaneously regulated in response to various types of stress. In addition, several studies have shown that the TFs involved in amino acid starvation and biofilm formation are involved in the expressions of both *rsd* and *rmf* genes [8,9].

Metal ions are essential for diverse cellular processes, such as photosynthesis, gluconeogenesis, glycolysis, signal transduction, stringent response, sporulation, and pathogenesis [10,11,12,13]. However, metal ions are toxic at high intracellular concentrations [14,15,16] and their levels are regulated by homeostatic mechanisms involving metal-responsive TFs [17,18,19,20,21]. Therefore, in this study, we investigated the involvement of metal-responsive TFs in regulating *rsd* and *rmf* gene expressions.

In stress caused by either a deficiency in or an excess of metal ions, we hypothesized that metal-responsive TFs might be involved in regulating the expression of *rsd* and *rmf* genes, which regulate the transcriptional and translational activities in the cell. Therefore, we examined the involvement of several metal-responsive TFs, namely BasR, CueR, CusR, Fur, KdpE, MntR, NhaR, PhoP, ZntR, ZraR, and Zur (Table 1). BasR and BasS form a typical two-component system, which functions as an iron–zinc-induced transcription regulator for a group of genes related to metal-responsive membrane structure modification and function regulation [22,23]. CueR and CusR regulate intracellular copper levels [16,24]. CueR is activated mainly under aerobic conditions, while CusR is activated under anaerobic conditions [25]. Fur regulates intracellular iron concentration and is essential for many processes, such as DNA synthesis and respiration [21,26]. The two-component KdpD/KdpE system regulates the transport of K^+^, the most abundant ion in the cell, with KdpD acting as the sensor kinase and KdpE acting as the response regulator [27,28]. MntR is associated with the regulation of intracellular Mn^2+^ concentration, glycogenesis, and oxidative stress [15,29]. NhaR participates in the maintenance of sodium concentration by regulating the expression of the membrane protein NhaA [30,31]. PhoP, in conjunction with PhoQ, functions as a two-component system facilitating Mg^2+^ transport [32,33], and ZntR, ZraR, and Zur aid in the regulation of intracellular Zn^2+^ concentration, an important component of many proteins [14,34,35,36,37,38].

Herein, we first examined whether metal-responsive TFs (listed in Table 1) bind to the promoter regions of the *rsd* and *rmf* genes. After that, we then investigated the effect of each TF gene-deficient strain on the expression of *rsd* and *rmf*. We believe that this study will be useful for understanding bacterial survival strategies because transcriptional and translational regulations in stress conditions, such as a deficiency in or an excess of metal ions, are essential survival mechanisms for bacteria, and our findings will provide new insights into infectious disease control, where long-term survival is an obstacle to countermeasures.

## 2. Results

### 2.1. Many Metal-Responsive TFs May Be Able to Bind to the Promoters of the rsd and rmf Genes

Two assays were used to examine whether metal-responsive TFs bind to the promoter regions of the *rsd* and *rmf* genes. First, the TFs used for in vitro assays were purified (Appendix A), and proteins other than His-MntR (Appendix A) and His-ZraR (Appendix A) were obtained without difficulty; however, only a small amount of His-MntR was obtained. In particular, several bands obtained during sodium dodecyl sulfate–polyacrylamide gel electrophoresis (SDS-PAGE) were slightly smaller than those representing the molecular weight of intact His-MntR, suggesting that the purified MntR may have been degraded. Moreover, two His-ZraR bands were obtained using SDS-PAGE, and since the His-tag was attached to the N-terminus of the protein, several amino acids in the C-terminal region may have been trimmed.

Figure 1 shows the electrophoresis patterns for examining TFs with *rsd* and *rmf* promoter-binding activity using the PS-TF screening system (see Section 4.4). The *rtcA* DNA probe is a reference DNA fragment corresponding to the open reading frame sequence of the *rtcA* gene, to which TFs are expected not to bind. Without the addition of a TF, each of the three DNA probes formed a single band against the estimated size, as shown in Figure 1A. However, if the added TF binds to the DNA probe, the band of the probe migrates upward from the position shown in Figure 1A. Dan (previous name: YgiP, a DNA-binding protein under anaerobic conditions) was employed as the negative control since it was found from a previous PS-TF screening [7] not to bind to either the *rsd* or *rmf* probes. In contrast, SdiA (a quorum-sensing system regulator) was found to bind to both *rsd* and *rmf* probes and was employed as the positive control. Similar to a previous study [7], Dan did not bind to either the *rsd* or *rmf* probes (Figure 1B), while SdiA was bound to both probes (Figure 1C). Table 1 shows the characteristics of transcription factors (TFs) targeted in this study. Figure 1D–N shows the electrophoresis patterns demonstrating the binding of various TFs and probes. Remarkably, CueR was clearly bound to both *rsd* and *rmf* probes, as shown in Figure 1E, whereas the others showed no binding. Most TFs examined did not bind to the probes, probably because of the lack of additional effectors for each of them. Considering this possibility, electrophoresis was performed by adding effectors (metal ions) to the TF and probe mixture. However, the effectors inhibited electrophoresis (Appendix A). Therefore, we designed an experimental method to verify the binding of TFs and DNA probes using a magnetic bead assay (see Figure 2, Section 4.5). In this assay, fluorescence can be detected if the fluorescein-4-isothiocyanate (FITC)-labeled DNA probe binds to the His-tagged TF in the solution eluted with imidazole (Figure 2E). Figure 3 shows the analyzed fluorescence results using a laser scanner, and the data without effectors can be compared with the PS-TF data in Figure 2. Figure 3A shows the results obtained after the addition of Dan and SdiA. The upper panel is a scanned image of the fluorescence emission state of the solution, and the lower panel is a graph normalized by the fluorescence intensity in the case of Dan, which is known not to emit fluorescence. SdiA binds to both probes and shows fluorescence. Figure 3B–L showed the results when metal-responsive TFs were added. BasR, Fur, KdpE, NhaR, PhoP, ZntR, ZraR, and Zur were bound to the *rsd* and *rmf* probes after adding the effector, as shown in Figure 3B,E,F,H–L, respectively, whereas in Figure 3C, although CueR binds to the *rsd* and *rmf* probes without the effector (CuSO_4_), its addition increases the extent of binding (Figure 1E). It is also worth noting that CusR and MntR (Figure 3D,G, respectively) did not show clear binding to the probes, even when the effector was added. The PS-TF screening and beads assay, which examine binding to the *rsd* and *rmf* promoters for target TFs [7], have been performed multiple times, and their results were reproduced in this study (Figure 1 and Figure 3). However, these data were obtained in vitro using His-tagged protein, which may bind non-specifically to DNA in the presence of metal ions [42]. Therefore, these results do not guarantee the binding or non-binding of each TF to the *rsd* and *rmf* promoters in vivo but only indicate the possibility of occurrence. For example, we looked for non-specific binding of His-tagged protein (Dan) to the *rsd* promoter by the addition of each effector and could not observe any such binding, as shown in Appendix A.

### 2.2. Genetic Defects in Some Metal-Responsive TFs Affect Rsd and Rmf Transcript Levels

The effect of gene deletions in each TF on *rsd* and *rmf* transcription levels was examined through quantitative PCR (qPCR) after reverse transcription. Each bacterial cell used in qPCR analysis was harvested 24 h after incubating the culture. Analysis of the qPCR results confirmed that the mRNA of each TF in the parental strain was transcribed under culture conditions (Appendix A). Gene deletion of each TF was confirmed by the lack of each transcription level in each TF gene-deficient strain harvested 24 h after incubating the culture in a similar experiment as Appendix A. Figure 4 shows the relative transcript levels of *rsd* (A) and *rmf* (B) measured using qPCR and normalized to those of the parental strain. When *basR*, *cusR*, *zraR*, and *zur* were deleted, the transcription level of *rsd* was comparable to that of the parental strain (Figure 4A). Since CusR does not bind to the *rsd* promoter, as shown in Figure 3, it was assumed that the transcription level of *rsd* did not change. Similarly, despite the binding of BasR, ZraR, and Zur to the *rsd* promoter, the transcription levels of *rsd* did not change in these gene-deficient strains (Figure 3). However, the transcription levels of *rsd* increased when the genes encoding other TFs were deleted. In addition, despite the lack of binding of MntR to the *rsd* promoter (Figure 3), the transcription levels of *rsd* appeared to increase in the mutant strain with an *mntR* deletion.

Figure 4B shows the relative transcription level of *rmf*. When *basR*, *cusR*, and *zur* were deleted, the transcription levels of *rmf* were comparable to those of the parental strain, and this behavior is similar to that observed for *rsd* transcription. However, the transcription levels of *rsd* decreased when the *fur* gene was deleted, and the deletion of genes encoding other TFs increased the transcription levels of *rmf*.

### 2.3. Genetic Defects in Some Metal-Responsive TFs Affect Rsd and Rmf Protein Levels

Next, we examined the effect of TF gene deletion on the expression of Rsd and RMF proteins. Figure 5 shows the results of Western blotting with antibodies against Rsd and RMF (as well as RpoA, YqjD, and RplB), performed using the cell extracts from each *E. coli* strain harvested 24 h after incubating the culture. YqjD is a membrane-binding protein expressed in the stationary phase that binds to ribosomes (Figure 5A) [43]. YqjD expression is regulated by the sigma factor RpoS (σ^s^), which is involved in stress response [43]. Rsd binds to the global regulator RpoD (σ^D^), allowing RpoS to bind to the RNA polymerase [2]. Thus, increased Rsd expression is expected to promote RpoS binding to RNA polymerase and enhance YqjD expression. RNA polymerase subunit alpha (RpoA), one of the proteins that make up RNA polymerase, normalizes the protein levels of Rsd and YqjD. 

Figure 5B,C shows the ratio of the band density of Rsd and YqjD to that of RpoA normalized by the parental strain (parent) data, respectively. As shown in Figure 5B, the deletion of *cueR*, *kdpE*, *mntR*, *nhaR*, *phoP*, or *zntR* genes increased Rsd expression, while the deletion of other TF genes did not significantly change the expression levels of Rsd. These results are similar to those of mRNA abundance analysis obtained through qPCR (Figure 4A). Moreover, the changes in YqjD expression in each TF-deficient strain (Figure 5C) were similar to the changes in Rsd expression (Figure 5B), thus supporting the speculation that *cueR*, *kdpE*, *mntR*, *nhaR*, *phoP*, or *zntR* gene expression is involved in Rsd expression. However, the deletion of the *fur* gene did not significantly change the expression levels of Rsd (Figure 5A) but increased that of YqjD (Figure 5B). Furthermore, *rsd* mRNA expression appeared to increase after *fur* gene deletion (Figure 4A), suggesting that *fur* is involved in *rsd* transcription.

Ribosomal protein L2 (RplB) is one of the core proteins that make up the 50S ribosomal subunit and normalizes the level of RMF. Figure 5D shows the results of Western blotting using the RMF antibody. Figure 5E shows the ratio of the RMF band density to the RplB band density normalized by the bar data. The expression of RMF was clearly increased by deletion of the *kdpE*, *mntR*, *nhaR*, *phoP*, *zntR*, or *zraR* genes and decreased by the loss of the *fur* gene, which is similar to the results of mRNA abundance analysis obtained using qPCR (Figure 4B).

### 2.4. Genetic Defects in Some Metal-Responsive TFs Affect 100S Ribosome Formation

RMF is a key factor in 100S ribosomal formation, and its behavior is directly related to the number of 100S ribosomes. Therefore, we examined their formation in each TF-deficient strain (Figure 6). Figure 6A–L shows the ribosome profiles obtained by analyzing the cell extracts obtained from sucrose density gradient centrifugation of each mutant strain cultured for 24 h. The 70S and 100S ribosomes were observed in all strains. Figure 6M shows the 70S to 100S ribosome abundance ratio (100S/70S ratio) for each strain, as calculated by the Systat software (Systat Software, Chicago, IL, USA) from the waveform of the ribosome profile. Genetic deletions of *cueR*, *kdpE*, *mntR*, *nhaR*, *phoP*, *zntR*, and *zraR* increased the number of 100S ribosomes formed as opposed to their parental strains. However, the number of 100S ribosomes formed by genetic deletions of *basR*, *cusR*, and *zur* was similar to that of the parental strains. Furthermore, the genetic deletion of *fur* actually reduced the number of 100S ribosomes formed. These results are similar to the data on RMF levels in each mutant strain (Figure 5D,E).

## 3. Discussion

In this study, we investigated the involvement of metal-responsive TFs in the expression of Rsd and RMF proteins, which regulate transcriptional and translational activities under stress. In vitro assays showed that most TFs bind the promoter regions of *rsd* and *rmf* genes following the addition of effectors (Figure 1 and Figure 3). However, CusR and MntR did not bind to the promoter regions of the *rsd* and *rmf* genes, even when an effector, acetyl phosphate (AcP) or MnCl_2_, was added (Figure 3D,G), which is likely due to the absence of these TF-binding sites in the range of the promoters examined. In addition, we measured the changes in *rsd* and *rmf* mRNA levels (Figure 4), Rsd and RMF protein levels (Figure 5), YqjD protein levels affected by Rsd (Figure 5A,C), and 100S ribosome formation affected by RMF (Figure 6) when the gene encoding each TF was deleted. We found that the deletion of *cueR*, *kdpE*, *mntR*, *nhaR*, *phoP*, and *zntR* clearly increased *rsd* mRNA levels (Figure 4A) as well as Rsd (Figure 5B) and YqjD protein levels (Figure 5C). However, as shown in Figure 1I and Figure 3G, the binding of MntR to the *rsd* promoter was not observed. The His-tagged MntR used in this assay was difficult to express or purify, as shown in Appendix A; therefore, we considered that it may have been non-degradable or insoluble. Nevertheless, although in vitro assays have not confirmed the binding of MntR to the *rsd* promoter region, we presumed that this TF is involved in the expression of Rsd and the above TFs.

Furthermore, the deletion of *kdpE*, *mntR*, *nhaR*, *phoP*, *zntR*, and *zraR* clearly increased *rmf* mRNA levels (Figure 4B), RMF protein levels (Figure 5E), and 100S ribosome formation (Figure 6M). Although *cueR* gene deletion does not increase RMF levels (Figure 5E), *rmf* mRNA and 100S ribosomal levels were increased (Figure 4B and Figure 6M, respectively). Since the increase in the 100S ribosome level indicates an increase in the RMF protein level, it is assumed that CueR is also involved in the expression of RMF as well as the above TFs. A series of experiments (Figure 1, Figure 4 and Figure 6) has shown that CusR does not participate in the expression of Rsd or RMF. Furthermore, CusR is activated under anaerobic conditions, while CueR is activated under aerobic conditions [25]. Since this experiment was performed under aerobic conditions, we considered that the involvement of CusR in the expression of the *rsd* and *rmf* genes may not be apparent. However, the deletion of the *fur* gene clearly reduced *rmf* mRNA (Figure 4B), RMF protein (Figure 5E), and 100S ribosome levels (Figure 6M), unlike other TFs.

Based on these results, we discuss the effect of changes in the concentration of several metal ions on the transcriptional and translational activities of *E. coli* cells. Deletion of *cueR*, *kdpE*, *mntR*, *nhaR*, *phoP*, *zntR*, and *zraR* increases Rsd, RMF, and YqjD expression along with 100S ribosome formation. This indicates that these TFs repress the expression of *rsd* and *rmf* genes by binding their promoter regions. Although some of these TFs are known to have consensus sequences for DNA binding [39], the determination of exact binding sites in the promoter regions of *rsd* and *rmf* genes is difficult and remains a future challenge. Furthermore, several studies have shown that most *E. coli* promoters carry binding sites for multiple TFs, with each factor monitoring different environmental conditions or metabolic states [44,45]. Of these TFs, CueR, MntR, NhaR, and ZntR bind when metal ions such as Cu^2+^, Mn^2+^, Na^+^, and Zn^2+^ are present at high concentrations in the cell. In contrast, KdpE, PhoP, and ZraR are phosphorylated by response regulators and bind to promoter regions when K^+^, Mg^2+^, and Zn^2+^ are present at low concentrations in the cell. Thus, high and low concentrations of specific metal ions may be implicated in the repression of Rsd and RMF proteins. These facts indicate that inadequate concentrations of these metal ions reduce transcriptional and translational activity and suppress cellular activity. Additionally, the deletion of the *fur* gene did not alter the expression levels of Rsd and YqjD but reduced RMF expression and 100S ribosome formation. Therefore, this finding indicates that the formation of 100S ribosomes is inhibited in the absence of Fe^2+^. Since iron ions are essential for many processes, such as DNA synthesis and respiration [21,26], bacteria in the iron-ion-deficient condition may not have the margin of forming 100S ribosomes for long-term survival. A similar trend was observed in previous studies with ArcA [7], a transcription factor involved in redox regulation under anoxic conditions. ArcA has been reported to be involved in iron transport by Fur under anaerobic conditions [46], suggesting that the regulation of iron homeostasis is closely linked to the regulation of translational activity.

Previous studies have shown that TFs involved in biofilm formation bind to the promoter regions of *rsd* and *rmf* in response to nutrient depletion and other factors, promoting the expression of these genes [7]. In this study, we demonstrated that the expression of *rsd* and *rmf* is suppressed by metal-responsive TFs when the intracellular metal ion concentration is not appropriately balanced. Thus, the expression of *rsd* and *rmf* is regulated positively or negatively in response to various environmental changes for the long-term survival of bacteria such as *E. coli*.

## 4. Materials and Methods

### 4.1. Bacterial Strains and Growth Conditions

*E. coli* strains from the ASKA clone library [47] of the *E. coli* Stock Center (National Bio-Resource Center, Shizuoka, Japan) were used for TF production. For TF overexpression, *E. coli* cells were grown in Luria–Bertani (LB) broth at 37 °C with shaking.

Mutant strains with one of the TF genes deleted were obtained from the Keio collection [48] of the *E. coli* Stock Center (National Bio-Resource Center, Shizuoka, Japan.) Mutant cells were grown at 37 °C with shaking at 160 rpm in medium E containing 2% polypeptone and 0.5% glucose [49] under aerobic conditions. Medium E contains MgSO_4_, citric acid, K_2_HPO_4_, and NaNH_4_HPO_4_, in which *E. coli* cells can efficiently form 100S ribosomes under stress conditions. Cell growth was monitored by measuring turbidity at 600 nm using an OD-Monitor (TAITEC, Saitama, Japan).

### 4.2. Purification of TFs

Thirteen TFs were prepared for in vitro assays. *E. coli* cells carrying each of the TF expression plasmids were grown in LB broth medium up to an OD_600_ of 0.6; after that, 1 mM isopropyl-beta-D-thiogalactopyranoside was added to induce TF expression. The cells were then harvested, suspended in a lysis buffer, and disrupted by sonication. After DNase I treatment, the cell lysates were incubated on ice for 3 h to digest genomic DNA and centrifuged to remove cell debris at 12,000 rpm (13,000× *g*) for 10 min at 4 °C. The cell lysates were passed through a HisTrap FF column (Cytiva, Tokyo, Japan) on an AKTA Prime system (Cytiva, Tokyo, Japan) pre-equilibrated with 20 mM PBS (pH 7.2), and the column was washed with the same buffer to remove unabsorbed proteins. The absorbed proteins were eluted with a linear gradient of 500 mM imidazole, from 0% to 100%, in 20 mM PBS (pH 7.2). The purity of each peak was determined using SDS-PAGE. The concentration of individual proteins was determined using the Protein Assay Rapid Kit (Wako Pure Chemical Industries, Osaka, Japan). 

### 4.3. Preparation of DNA Probes

FITC-labeled DNA probes were prepared by PCR amplification of the *rsd* (300 bp) and *rmf* (256 bp) promoter regions [7]. A 193 bp-long FITC-labeled probe, which is part of the open reading frame sequence of the *rtcA* gene, was prepared as a reference probe that is not expected to bind TFs. FITC emits fluorescence at approximately 520 nm upon excitation with light at approximately 495 nm. The fluorescence intensity was measured using a Typhoon FLA 9000 laser scanner (Cytiva, Tokyo, Japan).

### 4.4. PS-TF Screening System

A PS-TF screening system was employed to detect TFs with *rsd* and *rmf* promoter-binding activity [50]. The DNA probes (0.5 pmol) were mixed with each purified TF (20 pmol). After incubation at 37 °C for 20 min, the mixture was subjected to PAGE to detect DNA–protein complexes. The gels were analyzed using a Typhoon FLA 9000 laser scanner. 

### 4.5. Magnetic Bead Assays

Binding of the DNA probe to the TF was detected using magnetic beads (MagneHis Ni particle; Promega, Madison, WI, USA) for purification of His-tagged proteins (Figure 2). First, 30 µL of magnetic beads, 5 pmol of DNA probe, and 200 pmol of His-tagged TF were added to a total volume of 40 µL MagneHis Binding buffer, mixed well in a 1.5 mL tube (Figure 2A), and incubated at room temperature (20–24 °C) for 2 min (Figure 2B). The final concentrations of added effectors were as follows: 10 mM of AcP (for Dan, CusR, KdpE, PhoP, and ZraR), 0.5 mM of CuSO_4_ (for CueR), 0.025 mM of FeSO_4_ (for Fur), 0.2 mM of MnCl_2_ (for MntR), 100 mM of NaCl (for NhaR), and 0.5 mM of ZnSO_4_ (for ZntR and Zur). Notably, a DNA probe with an affinity for a TF could bind to the His-tagged TF-bound magnetic beads. Then, a magnet was used to attract and collect the magnetic beads (Figure 2C). The supernatant was carefully removed using a pipette, and the tube was removed from the magnetic stand. After that, 150 µL of MagneHis Wash Buffer was added to the tube and mixed well through pipetting. The tube was set on a magnetic stand for approximately 30 s to capture the magnetic beads, and the supernatant was removed using a pipette. This washing process was repeated twice for a total of three washes (Figure 2D). After the final wash, the tube was removed from the magnetic stand, and 100 µL of MagneHis Elution Buffer containing 500 mM of imidazole was added and mixed in well through pipetting. After incubation at room temperature (20–24 °C) for 1–2 min, the tube was again placed on a magnetic stand to capture the magnetic beads, and the supernatant containing the free DNA probe was removed using a pipette (Figure 2D). Finally, 100 µL of the eluate was transferred to a well on a plate, and fluorescence was analyzed using a Typhoon FLA 9000 laser scanner (Figure 2E).

### 4.6. Quantitative PCR (qPCR)

Total RNA was extracted from *E. coli* cells using NucleoSpin RNA Plus (Macherey-Nagel, Düren, Germany). cDNA was synthesized using the PrimeScript RT Reagent Kit (TaKaRa Bio Inc., Kusatsu, Japan). PCR assays were conducted in a Mic Real-Time PCR Cycler (Bio Molecular Systems, Upper Coomera, Australia) using SYBR Premix Ex Taq 2 (TaKaRa Bio Inc., Kusatsu, Japan). The number of PCR cycles required to obtain DNA within the linear amplification range from the amplification curve was determined. The copy numbers of the samples were obtained after quantitative amplification of the target gene. The threshold cycle (CT) values of the sample DNAs were normalized to the reference CT values obtained using the values of 16S rRNA. The relative quantity of each target mRNA was obtained using the 2^−ΔΔCT^ method.

### 4.7. Western Blot Analysis

Harvested *E. coli* cells were treated with lysozyme, and whole-cell extracts were prepared by sonication. Total cell proteins were fractionated using Tricine-SDS-PAGE on 15% gels [51] and transferred onto polyvinylidene difluoride (PVDF) membranes (Immobilon-FL transfer membrane; Merck, Darmstadt, Germany). The proteins Rsd, RMF, YqjD, RpoA, and RplB were detected on the membranes using rabbit polyclonal antibodies against Rsd, RMF, YqjD, RpoA, and RplB, respectively. Immunostained membranes with ECL substrate (Cytiva, Tokyo, Japan) were scanned using ImageQuant LAS 500 (Cytiva, Tokyo, Japan). The density of each band on the membranes was quantified using ImageJ software (https://imagej.nih.gov/ij/index.html (accessed on 24 August 2022)). The linearity of the quantification was confirmed through several experiments with different amounts of sample solution loaded on an electrophoresis gel.

### 4.8. Measurement of 100S Ribosome Level Using Sucrose Density Gradient Centrifugation

*E. coli* was grown in medium E, containing 2% polypeptone and 0.5% glucose. The pellets of *E. coli* cells harvested 24 h after inoculation were suspended in an association buffer (100 mM NH_4_ acetate, 15 mM magnesium acetate, 20 mM Tris-HCl [pH 7.6], and 6 mM 2-mercaptoethanol) and mixed with an equal volume of glass beads (212–300 µm; Merck, Darmstadt, Germany). The homogenate was centrifuged at 15,000 rpm for 10 min at 4 °C. The supernatant was layered on top of a 5–20% linear sucrose density gradient prepared in the association buffer and centrifuged in an SW41 Ti rotor (Beckman Coulter, Brea, CA, USA) at 41,000 rpm (288,000× *g*) for 2 h at 4 °C. After centrifugation, the absorbance of the sucrose gradient was measured at 260 nm using a UV-1800 spectrophotometer (Shimadzu, Kyoto, Japan), and the ribosome profile was drawn using UV-Prove software (Shimadzu, Kyoto, Japan). The ratio between 70S and 100S ribosomes was calculated for each peak using PeakFit software (Systat Software, Chicago, IL, USA) for peak separation analysis.

## 5. Conclusions

We have demonstrated that the metal-responsive TFs CueR, Fur, KdpE, MntR, NhaR, PhoP, ZntR, and ZraR are involved in the expression of *rsd* and *rmf* genes. These genes regulate transcription and translation under stressful environments caused by imbalanced metal ion concentrations. Expression of Rsd and RMF is suppressed at high concentrations of Cu^2+^, Mn^2+^, Na^+^, and Zn^2+^ and low concentrations of K^+^, Mg^2+^, and Zn^2+^. Furthermore, since *rmf* gene expression is reduced by the deletion of *fur* gene, we believe that the formation of 100S ribosome is inhibited in the absence of Fe^2+^. These results are important for understanding bacterial survival strategies under stress conditions and provide new insights into countermeasures for diseases in which stress tolerance is an issue. However, other TFs besides those addressed in this study may also be involved in the expression of *rsd* and *rmf* genes [7]. Thus, further elucidation of these additional TFs is needed to obtain a complete picture of bacterial survival strategies.

## Figures and Tables

**Figure 1 ijms-24-04717-f001:**
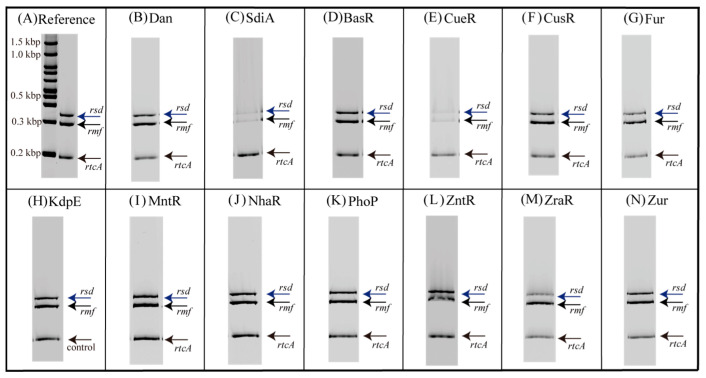
Promoter-specific transcription factor (TF) screening for metal-responsive TFs with *rsd* and *rmf* promoter-binding activity. Three fluorescein-4-isothiocyanate-labeled DNA probes (0.5 pmol each; specific for the 300 bp-long *rsd* promoter, the 256 bp-long *rmf* promoter, or the 193 bp-long *rtcA* internal DNA as a reference) were each mixed with 20 pmol of purified TFs (Appendix A) in 10 µL of DNA-binding buffer and incubated at 37 °C for 20 min. The DNA–protein mixtures were then directly subjected to PAGE to detect DNA–protein complexes. (**A**) No TF was added to the samples. (**B**) Dan, known not to bind either the *rsd* or *rmf* promoters, was added. (**C**) SdiA, known to bind both *rsd* and *rmf* promoters, was added. (**D**–**N**) Metal-responsive TFs were added.

**Figure 2 ijms-24-04717-f002:**
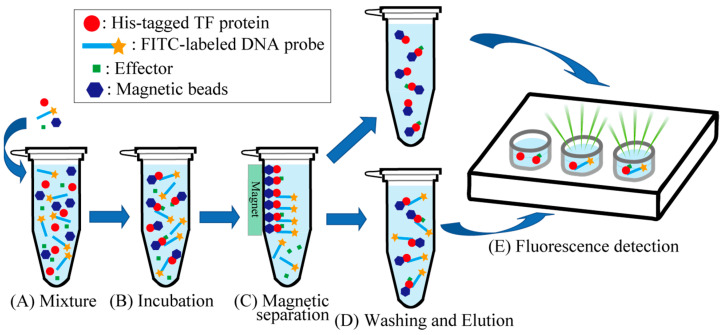
Magnetic bead assay for the detection of promoter DNA and transcription factor (TF) in the presence of metal ions. (**A**) Magnetic beads, a DNA probe, and a His-tagged TF (and effector) were added to MagneHis Binding Buffer. (**B**) The mixture was incubated at room temperature (20–24 °C) for 2 min. (**C**) The magnetic beads were attracted using a magnet. (**D**) The supernatant was removed using a pipette, and MagneHis Wash Buffer was added to the tube. MagneHis Elution Buffer was added after removing the supernatant. (**E**) The eluate was transferred to a plate, and the fluorescence of the solution was analyzed using a laser scanner.

**Figure 3 ijms-24-04717-f003:**
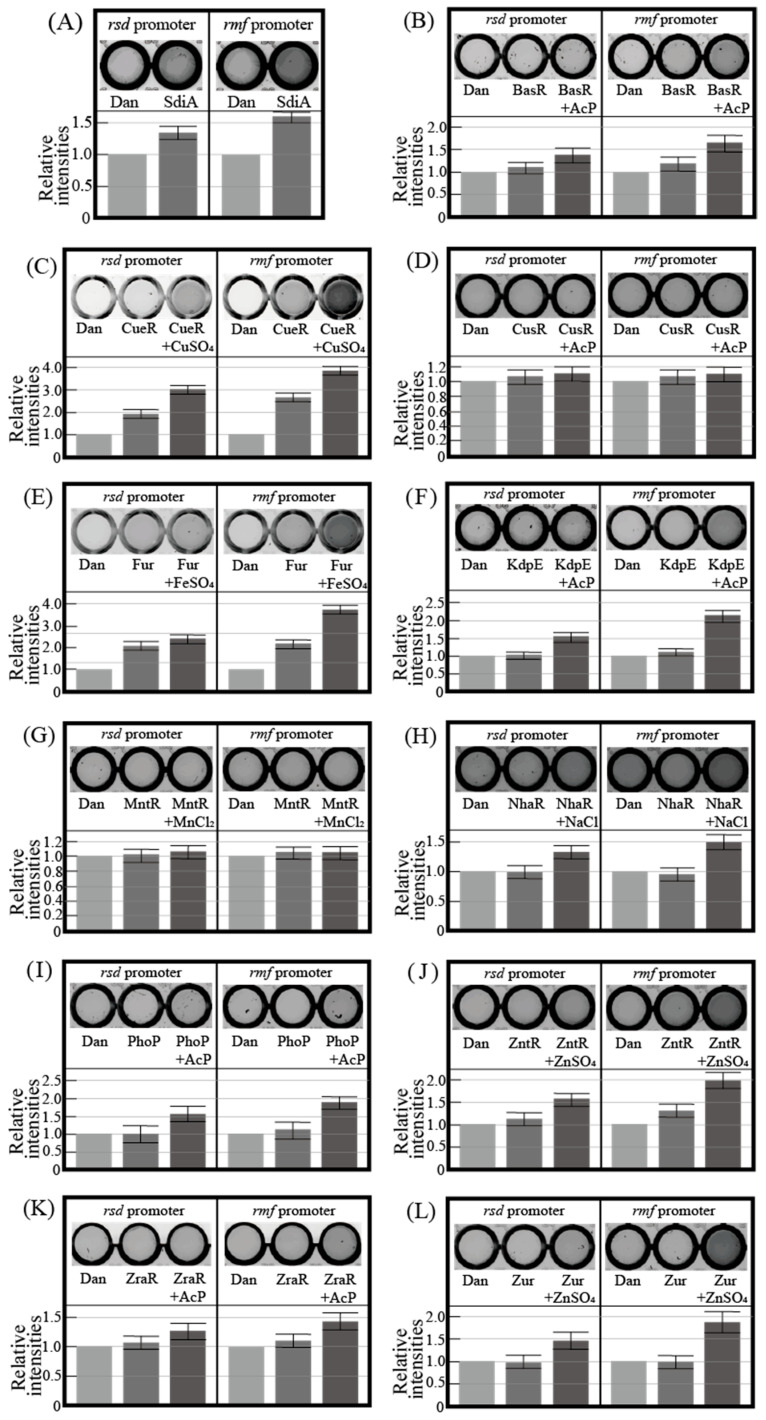
Fluorescence images of plate wells containing solutions obtained from the magnetic bead assay and the relative fluorescence intensities. Each upper panel indicates a scanned image, and each lower panel indicates a graph normalized against the fluorescence intensity for Dan. The scanned image is representative, and the error bars in the graph are the probable errors calculated from three experiments. (**A**) Dan or SdiA was mixed with *rsd* and *rmf* promoters. (**B**–**L**) Each transcription factor was mixed with promoters with or without the effector.

**Figure 4 ijms-24-04717-f004:**
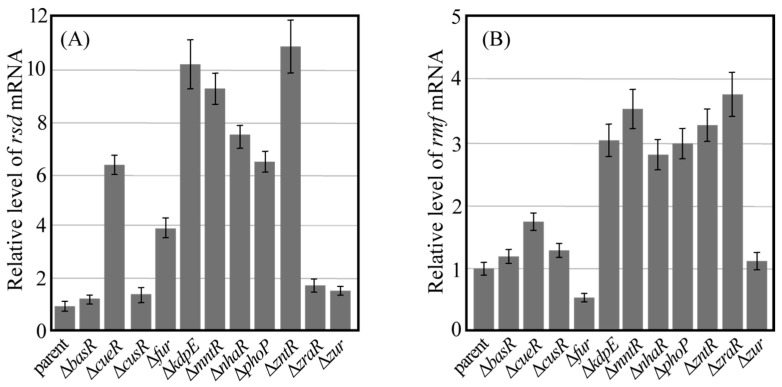
Amount of mRNA transcribed from *rsd* and *rmf* genes. The parental strain (parent) and each mutant strain lacking the transcription factor gene were harvested 24 h after inoculation. The amount of mRNA transcribed from the *rsd* (**A**) and *rmf* (**B**) genes in the cells was measured using qPCR. Values for each mutant were normalized against the value from their corresponding parent. Notably, each measurement was taken at least three times.

**Figure 5 ijms-24-04717-f005:**
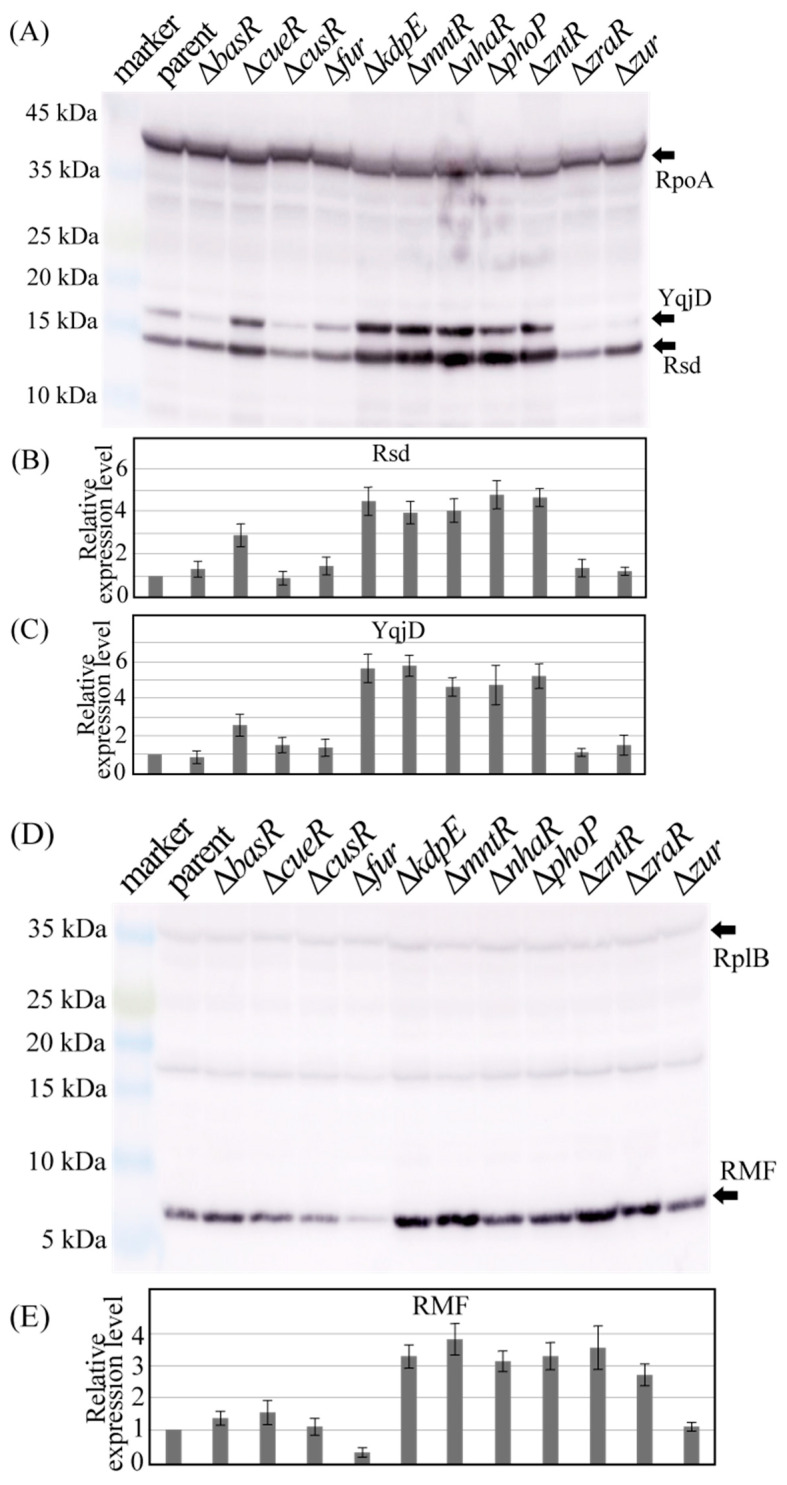
Protein levels of Rsd and RMF in each strain. The parental strain (parent) was harvested along with every mutant strain lacking the transcription factor gene 24 h after inoculation, and the cell extracts were analyzed using Western blotting. (**A**) Image of blotting membrane used for detecting Rsd and YqjD; RpoA was used as a reference. (**B**,**C**) Relative expression levels of Rsd and YqjD, respectively. The Rsd and YqjD band concentrations were divided by those of RpoA, and the values for each mutant were normalized against the value of their corresponding parent. (**D**) Western blot image of RMF; RplB was used as a reference. (**E**) Relative expression levels of RMF. The concentrations of RMF bands were divided by those of RplB, and the values for each mutant were normalized against the value of their corresponding parent. The images in (**A**,**D**) are representative, and the error bars in (**B**,**C**,**E**) are the probable errors calculated from three experiments.

**Figure 6 ijms-24-04717-f006:**
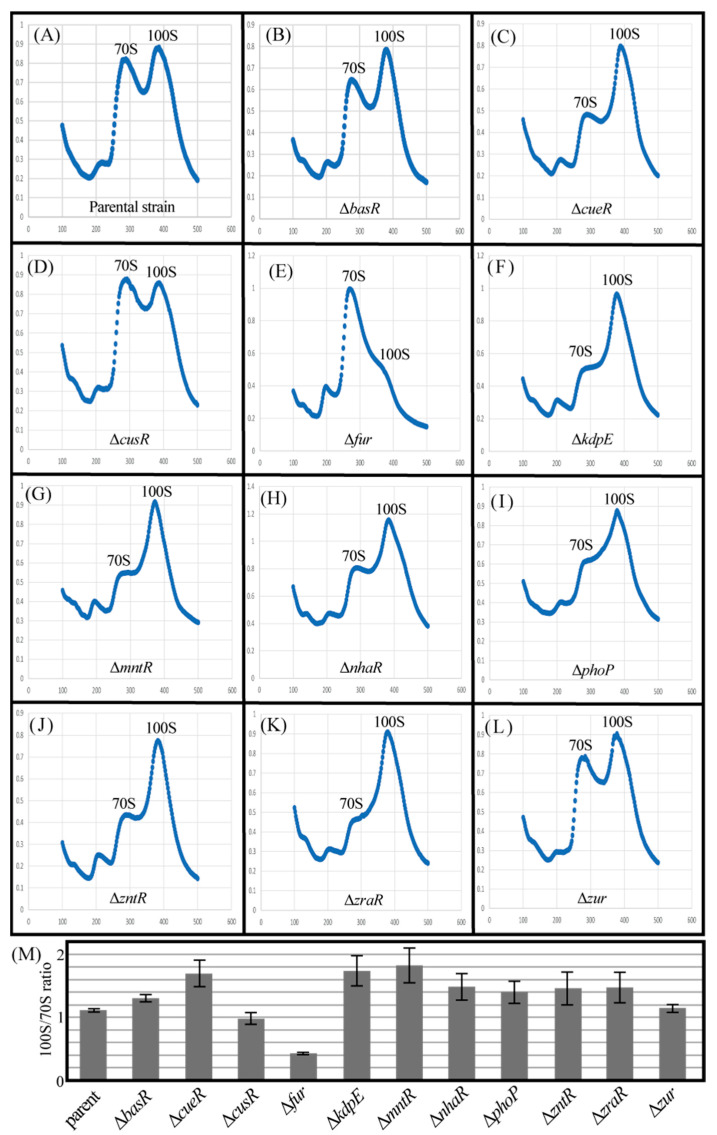
Influence of lack of metal-responsive TFs on the formation of 100S ribosome dimers. Each strain was harvested 24 h after inoculation. (**A**–**L**) The cell lysate of each strain was subjected to sucrose gradient centrifugation to monitor ribosome patterns. (**M**) For peak separation analysis, the level of 70S and 100S ribosomes was estimated using Systat software (Systat Software, Chicago, IL, USA). The quantitative ratios of the 70S and 100S ribosomes (100S/70S ratio) were calculated, and the values for each mutant were normalized against the value of their corresponding parent. Notably, the experiments were repeated at least three times.

**Table 1 ijms-24-04717-t001:** Characteristics of the targeted transcription factors (TFs).

TF	Mr(Da)	Family	Regulatory Function(s)	Effector	No. of Targets	Target TFs
BasR	25,031	OmpR	Biofilm formation and expression of outer membrane protein	TCS	13–17 [22]	PutA, CsgD
CueR	15,235	MerR	Copper efflux regulator	Cu^2+^	2 [39]	-
CusR	25,395	OmpR	Efflux of copper and silver	TCS	5 [40]	CusR, HprR
Fur	16,795	Fur	Ferric uptake regulator	Fe^2+^	132 [39]	FlhDC, Fur, MetJ, Nac, PurR, SoxS, SoxR
KdpE	25,362	OmpR	Potassium (K^+^) uptake	TCS	1 [41]	KdpE
MntR	17,640	DtxR	Manganese transport regulator	Mn^2+^	5 [39]	Dps
NhaR	34,284	LysR	Adaptation to Na^+^ and alkaline pH, and biofilm formation	Na^+^	3 [39]	NhaR
PhoP	25,535	OmpR	Two-component regulatory system with PhoQ	TCS	56 [39]	ArcA, GadE, GadW, PhoP, RstA, TreR
ZntR	16,179	MerR	Zn(II)-responsive regulator of zntA	Zn^2+^	1 [39]	-
ZraR	48,395	NtrC	Two-component regulatory system with ZraS	TCS	3 [39]	ZraR
Zur	19,254	Fur	Zinc uptake regulator	Zn^2+^	6 [39]	-

TCS: two-component system.

## Data Availability

No new data were created or analyzed in this study. Data sharing is not applicable to this article.

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
