# Peer review of "Metal-Responsive Transcription Factors Co-Regulate Anti-Sigma Factor (Rsd) and Ribosome Dimerization Factor Expression"

_ijms, 2023, doi:10.3390/ijms24054717_

Round 1

Reviewer 1 Report

The article “Metal-responsive transcription factors co-regulate anti-sigma factor (Rsd) and ribosome dimerization factor expression” by Yoshida et al. presents an interesting study on the regulation of two important regulatory factors: rmf and rsd. It is a relatively well performed study with several assays to support the role of the TFs tested to support their real involvement in regulating rmf and rsd. Nevertheless, a few more controls would be welcome, as well as added discussion, here are the specific comments.

P2, section 2.1: Mention that PS-TF can be compared to a gel shift assay (EMSA).

P3, line 103-104: I think part of the sentence is missing, I don't understand it (“Dan did not bind either…” presumably rmf and rms promoter, but specify).

P3: Line104: What is SdiA? Even if it is a control, a few words on it would be welcome.

P3, line 113: The His-Tag is still on TF, can it prevent the interaction on TF with its target? Or the otherway around, His is sometimes involved in nucleic acid binding sites (due to its potential positive charge), could it increase the probability of binding? According to (Paul et al. 2020, Protein Expression and Purification), His-Tag-dependent DNA binding can occur in some cases, especially in presence of some metals (see next comment for a control to circumvent that possibility).

P4, Figure 3: What is the threshold to say that a TF binds to the DNA? Some example at 1.5 compared to 1 for the negative control. Are the differences statistically significant (are there replicates of these?)? I see no error bars. Also, a negative control of Dan +added effector should be included (CuSO4, Acp, MnCl2, etc). Indeed, divalent cations have a significant impact on neutralization of the nucleic acids phosphate backbone charge. It might even be appropriate to choose one of the TFs that bind as an added control (for example SdiA) and use it for comparison with all the added effectors (perhaps in a suppl. Fig to avoid overloading fig3)

P6, line 158: Gene deletion of each TF was confirmed, how?

P7, line 194: RpsS should be replace by RpoS

P6, Figure 4: Suggestion, instead of Par., write “parent” it is about the same length as the KO gene denominations and will making the figure easier to read (without consulting the legend). Idem for figure 5.

P7, line 210: Figure 5B should be replace by Figure 5D.

P7, line 215: Relative to Figure 5E, why don't you mention the decrease of Rmf expression in the condition Δfur? This warrants more than the ~2 lines of text in the discussion (lines 294-296)

P7, line 226 / P9, Figure 6: Are the different between parental strain and deletions strain are statistically significant? Similarly, in fig 5, there are no error bars, is it because it was a single experiment?

P10, The effect of kdpE, mntR, nhaR, phoP, zntR, and zraR deletion have an impact, which could in principle mean either direct binding of the corresponding TF, or indirect regulation (if this TF affects other TFs for instance), the gel shifts help argue for direct binding, but this would also mean that numerous TFs bind the same promoter (especially since the study of ref #7 also demonstrates binding of many other TFs on these same two promoters). Therefore, additional discussion on the implication of the binding of so many TFs in the same promoter regions would be welcome, to compare with literature (is it common to have so many TFs binding a single promoter region; and if not, are there other examples that can be cited?); to discuss the implication of such results with regards to TF binding site consensus; as well as to discuss the biological implications for this two genes.

Overall, this manuscript definitely has merit and I hope the authors will be able to perform the few changes suggested.

Reviewer 3 Report

The study entitled “Methal-responsive transcription factors co-regulate anti-sigma factor (Rsd) and ribosome dimerization factor expression” by Hideji Yoshida et al. investigated the effect of several transcription factors on the expression of Rsd and RMF, which are transcriptional and translational regulators that work at the stational phase. The authors performed a series of assays, promoter binding assay, mRNA assay, western blotting, and 100S ribosome assay. These data support that some TFs, especially fur, significantly affect the expression of Rsd and RMF. The manuscript is written succinctly and the conclusions are supported by the data. I recommend publication after correcting some remaining issues listed below.

Major points

1.      Lane  418 in the conclusion, the authors wrote, “Furthermore, the reduction in rmf gene expression due to the deletion of fur gene indicates that the formation of 100S ribosomes is inhibited in the absence of Fe2+”. I feel that this is overstatement because the authors have not checked the formation of 100S ribosome in the absence of Fe2+.

Minor points

1.      It is kinder for readers if the authors explain more about Dan (e.g., about the function in the cell) when it is first mentioned around lane 102.

2.      The two sentences that start from Lane 102, “Similar to the previous study…” and “Characteristics of transcription factors …” were hard to understood. Probably, some words are missing.

3.      In some parts, the wordings are seemingly too decisive. I feel a more precise phrase would be appropriate. For example, the statement“CusR and MntR did not bind to the probes,” in Lane 124 is not precise because both fluorescent slightly higher than that with Dan (Fig. 3D), which means CusR and MntR bound to the DNA although the binding levels were negligible.

4.      Lane 159. It would be better to use “using qPCR after reverse transcription” instead of “using qPCR,” because it is strange to measure transcripts using qPCR.

Round 2

Reviewer 1 Report

The authors provided answers to my comments. I have a few minor comments left:

Line 140-141: authors write: “…For example, the non-specific binding of His-tagged protein (Dan) to the rsd promoter by the addition of each effector was confirmed, as shown in Figure S3.”

Yet, figure S3 shows no binding of Dan to DNA (whether effectors are present or not).

Perhaps the authors meant: “…For example, we looked for non-specific binding of His-tagged protein (Dan) to the rsd promoter by the addition of each effector and could not observe any such binding, as shown in Figure S3.

Lines 334-336: The added sentence needs to be clarified: "Since iron ions are essential for many processes, such as DNA synthesis and respiration [21,26], they may not have the margin of forming 100S ribosomes for long-term survival." what does “they” refer to.

Also, the previous sentence: “…Therefore, this finding indicates that the formation of 100S ribosomes is inhibited in the absence of Fe2+…” makes sense when considering only the fur-KO strain results (i.e. in absence of fur, there is less RMF and les 100S, which can be considered roughly equivalent to the presence of fur, but in absence of iron). However, how do you reconcile this result with results in Fig3 where you have clear binding of fur+iron to the rmf promoter? This needs to be discussed. 

Author Response

Our manuscript has benefited greatly from your insightful suggestions. We thank you for your thoughtful suggestions and insights. The manuscript has been rechecked, and the necessary changes have been made in blue fonts in accordance with your suggestions. The responses to all comments are provided below.

Line 140-141: authors write: “…For example, the non-specific binding of His-tagged protein (Dan) to the rsd promoter by the addition of each effector was confirmed, as shown in Figure S3.” Yet, figure S3 shows no binding of Dan to DNA (whether effectors are present or not). Perhaps the authors meant: “…For example, we looked for non-specific binding of His-tagged protein (Dan) to the rsd promoter by the addition of each effector and could not observe any such binding, as shown in Figure S3.”

Response: We appreciate your comment. The correction has been made accordingly on lines 166-168.

Lines 334-336: The added sentence needs to be clarified: "Since iron ions are essential for many processes, such as DNA synthesis and respiration [21,26], they may not have the margin of forming 100S ribosomes for long-term survival." what does “they” refer to.

Response: We appreciate your insightful comment. The correction has been made accordingly on lines 335-336.

Also, the previous sentence: “…Therefore, this finding indicates that the formation of 100S ribosomes is inhibited in the absence of Fe2+…” makes sense when considering only the fur-KO strain results (i.e. in absence of fur, there is less RMF and les 100S, which can be considered roughly equivalent to the presence of fur, but in absence of iron). However, how do you reconcile this result with results in Fig3 where you have clear binding of fur+iron to the rmf promoter? This needs to be discussed.

Response: Thank you for your valuable comment. As you instructed, more explanations (and a reference paper) have been added on lines 337-341.